# Dopamine promotes aggression in mice via ventral tegmental area to lateral septum projections

Darshini Mahadevia[1,2,5], Rinki Saha[1,2,5], Alessia Manganaro [3], Nao Chuhma[2,4], Annette Ziolkowski-Blake[2], Ashlea A. Morgan[1,2], Dani Dumitriu [1,3], Stephen Rayport [2,4] & Mark S. Ansorge [1,2]✉

Septal-hypothalamic neuronal activity centrally mediates aggressive behavior and dopamine system hyperactivity is associated with elevated aggression. However, the causal role of dopamine in aggression and its target circuit mechanisms are largely unknown. To address this knowledge gap, we studied the modulatory role of the population- and projection-specific dopamine function in a murine model of aggressive behavior. We find that terminal activity of ventral tegmental area (VTA) dopaminergic neurons selectively projecting to the lateral septum (LS) is sufficient for promoting aggression and necessary for establishing baseline aggression. Within the LS, dopamine acts on D2-receptors to inhibit GABAergic neurons, and septal D2-signaling is necessary for VTA dopaminergic activity to promote aggression. Collectively, our data reveal a powerful modulatory influence of dopaminergic synaptic input on LS function and aggression, effectively linking the clinically pertinent hyper-dopaminergic model of aggression with the classic septal-hypothalamic aggression axis.

[1] Department of Psychiatry, Division of Developmental Neuroscience, Columbia University, New York, NY 10032, USA. [2] New York State Psychiatric Institute, New York, NY 10032, USA. [3] Department of Pediatrics, Division for Child and Adolescent Health, Columbia University Irving Medical Center, New York, NY 10032, USA. [4] Department of Psychiatry, Division of Molecular Therapeutics, Columbia University, New York, NY 10032, USA. [5]These authors contributed equally: Darshini Mahadevia, Rinki Saha. ✉email: ma2362@cumc.columbia.edu

Aggression is a complex social behavior that manifests across species from flies to humans[1,2]. It is defined as an overt act that can occur when the interest of two or more individuals conflict[3]. Classic lesion and stimulation studies have identified the septal-hypothalamic pathway as critical in controlling aggression[4–6]. Specifically, stimulation of the ventral-lateral subdivision of the ventral-medial hypothalamus (VMHvl)[7] and lesions of the LS both increase aggression across species[8–11]. GABAergic neurons in the LS project to the VMHvl, thus establishing top-down inhibition in this septal-hypothalamic aggression axis. Optogenetic experiments in mice validated this model, demonstrating that activation of GABAergic LS neurons projecting to the VMHvl inhibit aggression[6].

Monoamines are recognized for their capacity to modulate aggressive behavior[12]. While a subpopulation of serotonergic neurons that execute such influence has been identified[13,14], less is understood about the specific role of the dopaminergic neurons in controlling aggression. In humans, levels of dopamine (DA) metabolites in cerebrospinal fluid are positively correlated with psychopathic behavior[15]. Mutations in the gene coding for catechol-O-methyltransferase (COMT), responsible for the degradation of catecholamines including DA[16], have been linked to aggression[17]. Furthermore, D2 receptor antagonists are currently used as first-line treatment in managing pathological aggression, indicating a causal role for high DA in aggression[18,19]. In animals, the release of DA in the nucleus accumbens (NAc) is associated with increased aggression[20]. Mice lacking the dopamine transporter (DAT) exhibit a hyper-DAergic tone that correlates with increased aggression following mild social contact[21]. Furthermore, optogenetic activation of VTA DA neurons drives isolation-induced aggression[22].

Building on this initial insight, we here investigated the modulatory role of population-specific and projection-specific DA function in mice, using optogenetics to manipulate DAergic neuronal activities and analyze behavior and post-synaptic electrophysiological responses. We furthermore paired retrograde tracing with brain-wide imaging analysis to map DAergic projection pathways. Our results identify a fundamental role for a subpopulation of VTA DAergic neurons that selectively project to the LS in orchestrating aggressive behavior, and highlight D2 receptors on GABAergic neurons in the LS as targets for drugs that ameliorate pathological aggression.

## Results

**Optogenetic activation of DAergic neurons in the VTA but not in the substantia nigra pars compacta promotes aggression.** To begin to elucidate which DAergic neurons exert pro-aggressive influence over behavior, we optogenetically stimulated key DA circuit nodes and examined their role in behavior. First, we compared the effect of VTA versus substantia nigra pars compacta (SNc) DAergic neuron activation on aggressive behavior (Fig. 1a). For this experiment, we used DAT[IRESCre]; Ai32 mice, which conditionally express channelrhodopsin-2-EYFP (ChR2-EYFP) driven by the DAT promoter[23,24]. At 2 months of age, male mice were implanted with fiberoptic ferrules in the VTA or SNc. At 4 months of age, we stimulated and tested mice in isolation-induced aggression (Fig. 1a). We assessed aggression behavior in pairs of male mice where one mouse expressed ChR2 in DA neurons (carrying the DAT[IRESCre] and one Ai32 allele), while the other mouse did not (carrying only the Ai32 or the DAT[IRESCre] allele). In replication of our previous report[22], we found that optogenetic stimulation (473 nm, 20 Hz, 10 ms pulse duration) of VTA DAergic neurons significantly increased aggression, as demonstrated by an increase in time spent fighting (Fig. 1b–d) and decrease in the latency to tail-rattle, mount, and

bite (Supplementary Fig. 1a–c). Similar aggression patterns were observed against female mice (Supplementary Fig. 2a–e). Optogenetic stimulation of SNc DAergic neurons did not impact time (Fig. 1e–g) or latency (Supplementary Fig. 1d–f) measures of aggression, demonstrating a selective role for VTA DAergic neurons in promoting aggression.

**Lateral septum and nucleus accumbens receive distinct DAergic projections from the VTA.** DAT-positive VTA DAergic neurons project to the NAc, where increased DA release has been measured during aggressive behavior[11]. However, VTA DAergic neurons might also innervate other brain regions that are thought to modulate aggressive behavior, including the lateral septum (LS)[5,6] and the VMHvl[7]. Using Dat[IRESCre]; Ai32 mice for axonal tracing, we found prominent DAergic input into the NAc and LS (Supplementary Fig. 3a, b), but not the VMHvl (Supplementary Fig. 3c). The pattern of innervation in the LS formed a crescent band, matching the distribution of tyrosine hydroxylase (TH) immunoreactivity[25,26]. We termed this anatomically defined septal region medial crescent lateral septum (MCLS).

To investigate whether the DA projections in the MCLS originate from the VTA, we injected a retrograde Cre-dependent HSV virus conditionally expressing GCaMP (hEF1α-LS1L-GCaMP6s) into the LS (Fig. 2a). To anatomically compare the VTA DAergic neuronal population projecting to the LS with the population projecting to the NAc and to test for mutual collateralization, in separate mice we injected hEF1α-LS1L-GCaMP6s into the NAc (Fig. 2f). Four weeks after injection, brains were processed for double immunolabelling of TH and GFP (the latter to enhance the GFP component of the GCaMP molecule). Following brain-wide imaging (Supplementary Fig. 4a, b), we conducted a quantitative analysis of axonal projections and cell bodies, using the *Wholebrain* software[27] registration function in RStudio and custom ImageJ macro scripts[28]. In LS injected animals, we detected a stream of retrogradely-labeled somas located in the VTA close to the midline (Fig. 2b, c, k, l) and in the periaqueductal gray (PAG) (Supplementary Fig. 10a–c and Supplementary Movie 1). The VTA-to-LS DAergic axonal projections formed a dense crescent structure in the MCLS (Fig. 2d). In NAc injected animals, we also detected a stream of retrogradely-labeled somas located in the VTA and PAG areas, but this stream was more lateral and less dense compared to LS-injected mice (Fig. 2g, h, k, l). Because GCaMP diffuses freely within the axonal lumen of neurons in which it recombines, we also assessed for collateralization. We did not detect GFP-positive axons in the LS of NAc injected mice (Fig. 2i), or in the NAc of LS injected mice (Fig. 2e), demonstrating that NAc and LS projecting DAergic neurons are different populations (Fig. 2o).

**DAergic input into the LS, but not the NAc, promotes aggression.** Next, we directly tested if DAergic projections to the LS or the NAc modulate aggression, with optogenetic stimulation in the MCLS or NAc in DAT[IRESCre]; Ai32 mice (Fig. 3a, f). We assessed aggression behavior in pairs of male mice where one mouse expressed ChR2 in DA neurons (carrying the DATIRESCre and two Ai32 alleles), while the other mouse did not (carrying only two Ai32 alleles). Optogenetic stimulation in the MCLS of ChR2 expressing mice (Supplementary Figs. 3b and 5) significantly increases time spent fighting (Fig. 3b, c) and decreased latency to attack (Supplementary Fig. 6a–c). Similar aggression patterns were observed against female mice as well (Supplementary Fig. 7a–e). Activation of DAergic terminals in the NAc (Supplementary Figs. 3a and 8) produced no significant difference in levels of aggression (Fig. 3g, h and Supplementary Fig. 6d–f). To investigate general motor activity, we stimulated both cohorts of mice in the open field and examined locomotor

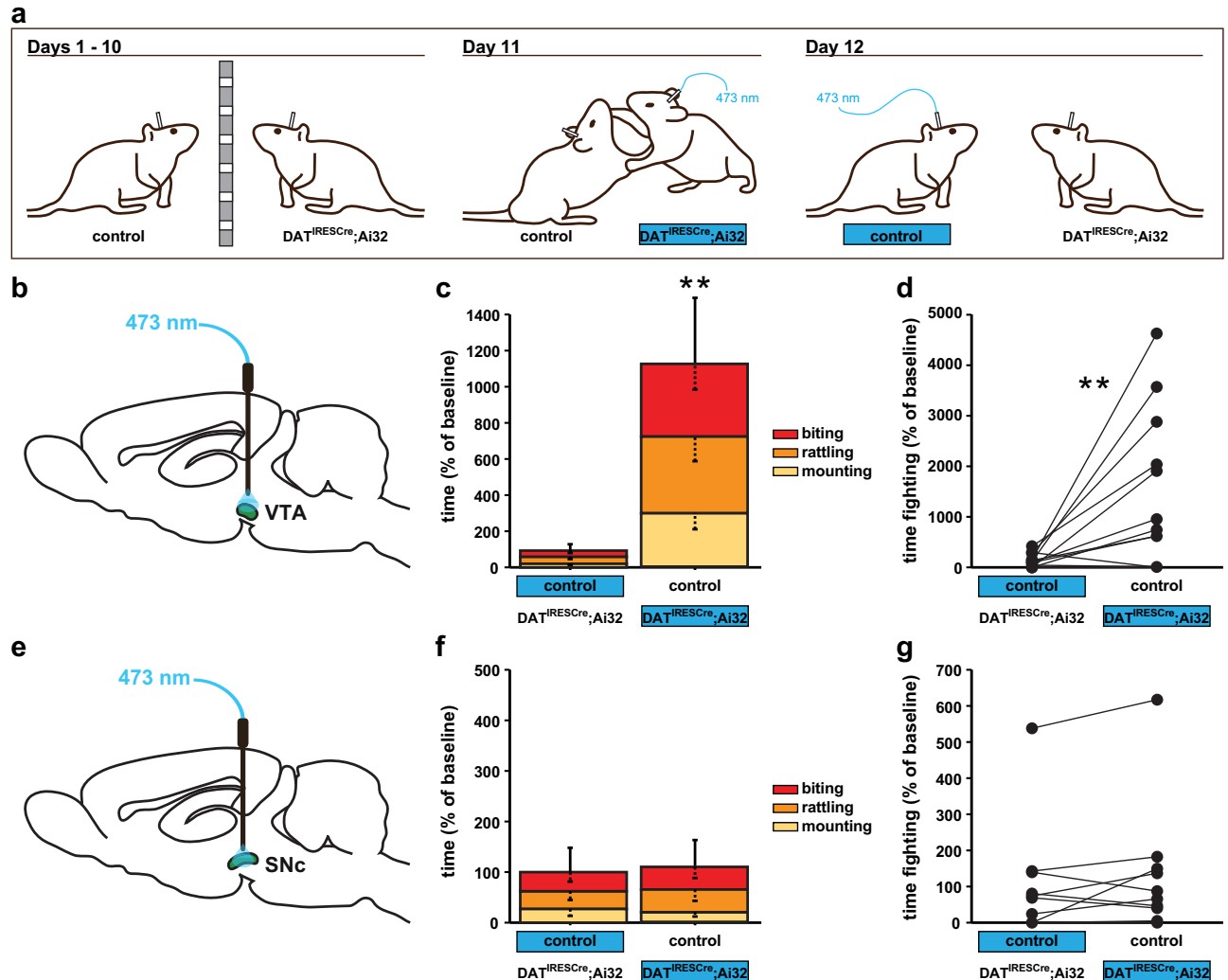

**Fig. 1 Optogenetic activation of DAergic VTA, but not SNc, neurons increases aggression. a** Schematic of the isolation-induced aggression experimental design between males. After separation for 10 days, mice underwent a stimulation protocol, counterbalanced for genotype on Day 11 and 12. Effects of optogenetic activation of the VTA (**b–d**) or SNc (**e–g**) on aggression. **b** The schematic diagram for stimulating VTA DA neurons in vivo. **c**, **d** Heightened aggressive behavior, measured as increased time fighting, was observed in pairs when DAT[IRESCre]; Ai32 mutant mice were stimulated (blue). **e** The schematic diagram for stimulating SNc DA release in vivo. **f**, **g** No effect of SNc DA stimulation was detected. *$P < 0.05$; **$P < 0.01$ compared with their respective controls; mean ± SEM; $n = 16$ (**c**, **d**) and 10 (**f**) pairs.

activity. We observed no increase in locomotor activity during optogenetic stimulation in either genotype in the MCLS cohort (Fig. 3d), while hyperactivity during optogenetic stimulation in ChR expressing mice was observed in the NAc cohort (Fig. 3i). Because DA plays a prominent role in motivation[29], we next tested animals in real-time place preference. While the NAc cohort showed a significant preference for the side associated with stimulation as expected (Fig. 3j), the LS cohort showed no real-time place preference (Fig. 3e). Finally, to test for off-target effects by optogenetic light diffusing from the MCLS to the dorsal striatum (DS), we activated DAergic input in the DS directly (Supplementary Figs. 3d and 9a) and saw no effect on aggressive behavior (Supplementary Fig. 9a–f). Together, these data reveal a dissociation between DAergic input to the NAc and LS and their respective roles in locomotion, reward, and aggressive behavior.

**Optogenetic activation of VTA-to-LS DAergic fibers increases aggression**. In the preceding experiment, we used DAT[IRESCre]

mice to conditionally drive ChR2 expression in the Ai32 line. This genetic strategy leads to the indelible expression of ChR2 in all DAT positive neurons. Indeed, we detected ChR2-positive cell bodies along an anterior-to-posterior stream ranging from the VTA to the PAG (Supplementary Fig. 10a–c). To specifically interrogate the VTA-to-LS pathway we adopted a viral approach: we injected male DAT[cre] mice with AAV5-ef1α-DIO-hChR2(H134R)-EYFP or AAV5-EF1a-DIO-EYFP control virus into the VTA to selectively express ChR2 in VTA DAergic neurons (Fig. 4b, c), and implanted a fiber optic ferrule above the MCLS (Fig. 4a and Supplementary Fig. 11a, b). Four weeks after viral injection and optic fiber implantation, animals were behaviorally tested. The same optogenetic stimulation (473 nm, 20 Hz, 10 ms pulse duration) in the MCLS significantly increased total fight time (Fig. 4d, e and Supplementary Fig. 12a–c). In the open field test, we detected a slight increase in locomotion during optogenetic stimulation (Fig. 4f). In real-time place preference, we did not detect an effect of optogenetic stimulation (Fig. 4g, h). Histological analysis confirmed no ChR2 or eArch expression in the PAG (Supplementary Fig. 13a–c), demonstrating that the viral

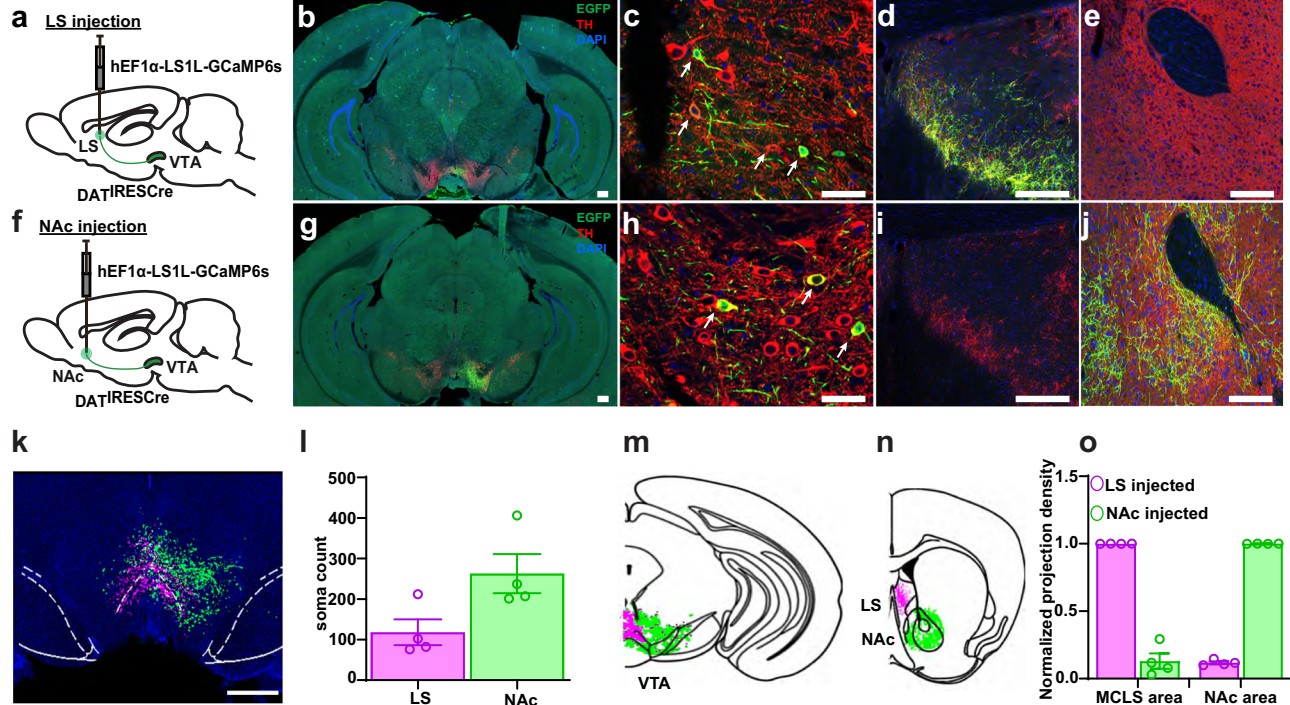

**Fig. 2 Retrograde labeling of DAergic VTA→LS and VTA→NAc projection neurons. a, f** Schematic diagram of retrograde hEF1α-LS1L-GCaMP6s injections into the LS and NAc of DAT[IRESCre] mice. **b–e, g–k** Confocal images from hEF1α-LS1L-GCaMP6s injected DAT[IRESCre] mice anti-GFP (green), anti-TH (red), and DAPI, (blue). **b, g** Whole section images containing the VTA. **c, h** Zoomed in confocal images of retrogradely labeled neurons in the VTA, arrows indicate double labeled cells. **d** Colocalization of GFP positive and TH positive fibers at the LS injection site. **e** Absence of GFP positive fibers in the NAc of LS injected mice. **i** Absence of GFP positive fibers in the LS of NAc injected mice. **j** Colocalization of GFP positive and TH positive fibers at the NAc injection site. **k** 2D reconstruction of the aggregate retrogradely labeled VTA-to-LS neurons (magenta) and VTA-to-NAc neurons (green) from four mice injected into the LS and four mice injected into the NAc. Soma locations were superimposed on one DAPI coronal image and overlaid with the outlines (white dashed lines) of the VTA brain areas from the Allen Mouse Brain Atlas. **l** VTA soma counts for LS and NAc injected animals. **m, n** Schematic of VTA-to-LS and VTA-to-NAc projection pathways. Aggregate of segmented somas (**m**) and terminals (**n**) for VTA→LS neurons (magenta) and VTA→NAc neurons (green). Coronal cartoon prepared based on Allen Brain Atlas. **o** Axonal projection density for LS and NAc injected brains, normalized by density at injection site. Scale bars: 300 μm (**b, g**), 200 μm (**d, e, i, j**), 50 μm (**c, h**), 1 mm (**k**). Quantification presented in mean ± SEM; n = 4 in each group.

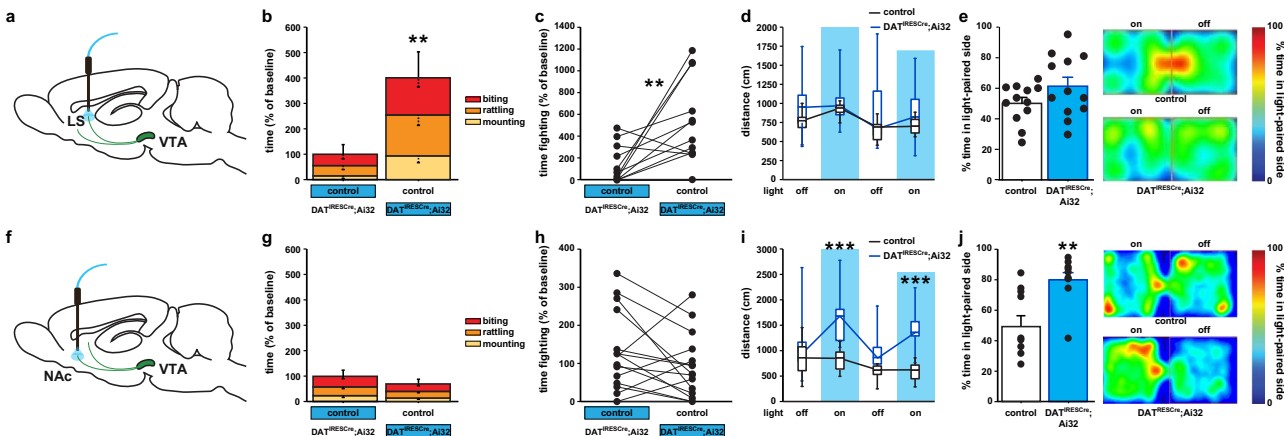

**Fig. 3 DAergic projections to the LS, not NAc, promote aggression. a, f** Schematic diagrams for terminal stimulation in vivo. **b, c** Heightened aggressive behavior was observed in pairs when DAT[IRESCre]; Ai32 mutant mice were stimulated (blue) in LS DA terminals. No effect of LS stimulation was detected in locomotor activity in the **d** open field or **e** real-time place preference (RTPP). **g, h** Aggressive behavior was not altered with NAc DA terminal stimulation (blue). **i** DA terminal stimulation in the NAc increased locomotor activity in the open field, during alternating 3 min off and on bouts of 20 Hz stimulation only in DAT[IRESCre]; Ai32 mutant mice. **j** The percentage of the time, over a 20 min session, spent in the stimulated zone during RTPP was significantly increased following stimulation in mutants. Also shown representative heat maps of the time spent in the stimulated zone of the chamber during RTPP. **P < 0.01; ***P < 0.001 compared with their respective controls; mean ± SEM; n = 16 pairs (**b, c**), 8–9/group (**d**), 12/group (**e**), 20 pairs (**g, h**) and 20/group (**i, j**), box plots represent median and IQR and whiskers extend to maximum and minimum values (**d, i**).

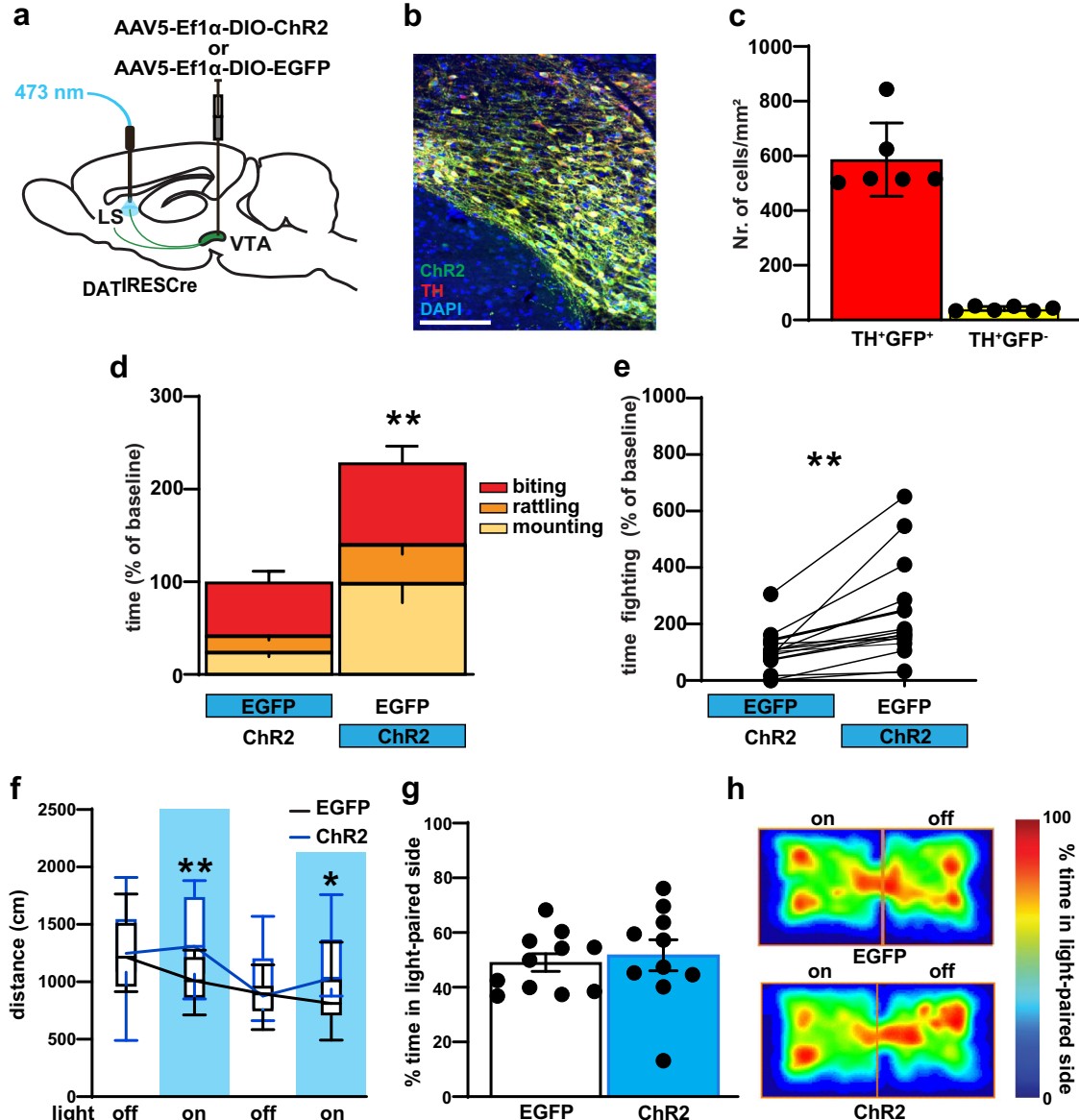

**Fig. 4 Optogenetic activation of VTA→LS DAergic terminals leads to increase aggression. a** Schematic diagrams for viral injection in DAT[IREScre] and terminal blue light stimulation in vivo. **b** Representative images of cell type-specific AAV-DIO-ChR2-eYFP expression (green) in VTA DA neurons (red), with colocalized neurons in yellow, Scale bar, 100 μm. **c** Quantification showed highly significant ChR2 expression in the TH[+] cell. **d, e** The increased aggressive response was observed in pairs when DAT[IREScre] mice were stimulated (blue) in LS DA terminals. **f** LS DAergic terminal stimulation increased locomotor activity in the open field, during alternating 3 min off and on bouts of 20 Hz stimulation only in DAT[IRESCre] ChR2 expressing mice. Vertical blue shading indicates optogenetic activation of ChR2. **g** During the RTPP task, percentage of time, over a 20 min session, spent in the stimulated zone was not significantly different in EYFP and ChR2 expressing mice. **h** The same result also displayed in the representative heat maps of the time spent in the stimulated zone of the chamber during RTPP. $P < 0.01$; ***$P < 0.001$ compared with their respective controls; mean ± SEM; $n = 15$ pairs, Box plots represent median and IQR and whiskers extend to maximum and minimum values (**f**). (Supplementary Data 1 for detailed statistics).

approach indeed specifically targets the DAergic VTA-to-LS pathway. Together these data demonstrate that neurotransmitter release from axon terminals of VTA-to-LS projecting neurons is sufficient to increase aggression without altering DA mediated reward preference.

**Axonal DA release in the LS induces inhibitory postsynaptic potentials in LS GABAergic neurons via D2 receptor activation.** In the LS, 85% of the neurons are GABAergic[6], and *Drd2* expression is higher than *Drd1* expression[30]. We therefore hypothesized that DA would act to hyperpolarize LS GABAergic projection neurons via D2 receptors and thereby reduce downstream

GABAergic tone. To test this hypothesis, we first evaluated *Drd2* and *GAD1* expression in the LS using RNAscope in situ hybridization. In *GAD1* positive cells, we found 82% colocalization with *Drd2*, and 95% of *Drd2* positive cells were also *GAD1* positive (Fig. 5a, b and Supplementary Fig. 14a). To evaluate DA neuron signaling to LS neurons, we next recorded from pyramidal-shaped neurons (located dorsal of the fornix) in acute brain slice preparations of the LS from DAT[IRESCre]; Ai32 mice. Many neurons were not spontaneously active (12 among 61 neurons were active; 3.3 ± 0.7 Hz) (Fig. 5c, left), but depolarizing current injection ($n = 14$ cells, 50 pA) caused firing (burst firing: $n = 11$ cells, tonic firing: $n = 3$ cells) (Fig. 5c, right). This finding suggests that firing of the LS neurons in vivo is mostly driven by excitatory synaptic inputs, not by intrinsic pacemaker firing.

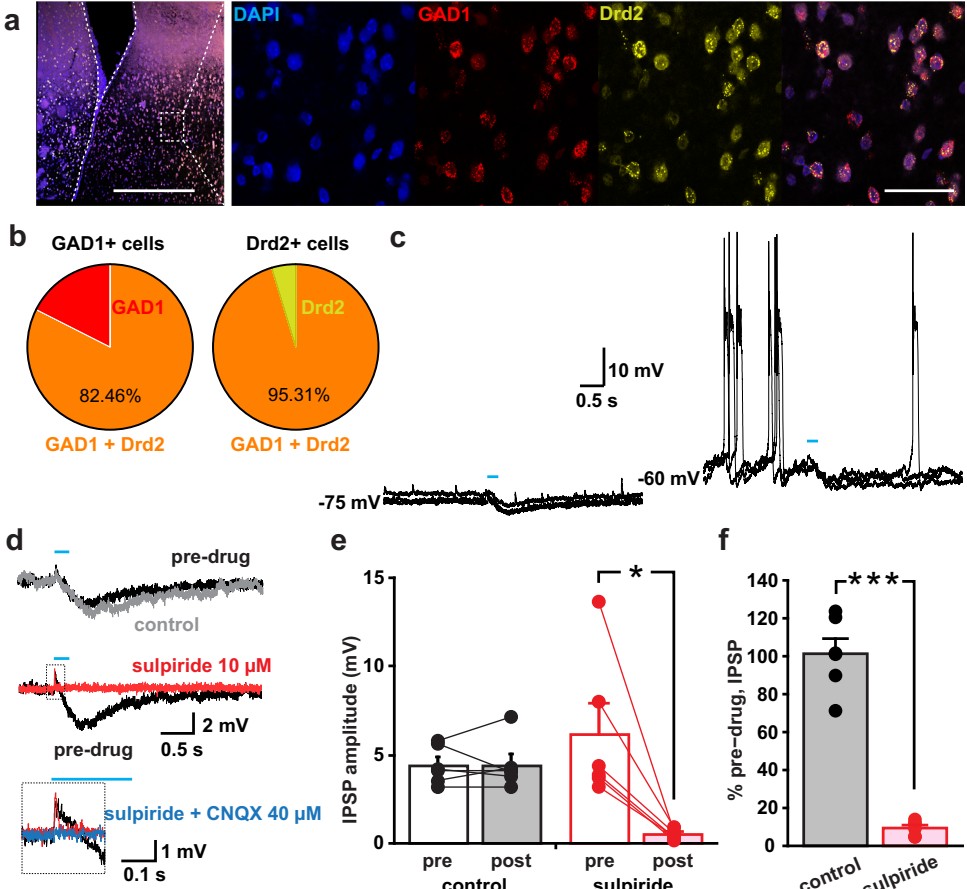

**Fig. 5 DA acts to hyperpolarize LS projection neurons via D2 receptor activation. a** Representative RNAscope image displaying colocalization of GAD1 (red) and Drd2 (yellow) mRNA in the LS; scale bars (500 and 50 μm). **b** The pie diagrams show high GAD1 & Drd2 colocalization in the LS; 82.46% of GAD1 positive neurons were also positive for Drd2 and 95.31% of Drd2 positive neurons were also positive for GAD1. **c** Sample traces of IPSPs without current injection (at −75 mV, left) and with the current injection to make the neuron fire (at −60 mV, right). Three traces are superimposed. Blue lines indicate the timing of train photo-stimulation (five pulses at 20 Hz). **d** Sample traces of IPSPs without depolarizing current injection before drug application (pre-drug), after 5 min application of control solution without drugs (control), and after 5 min application of sulpiride. The small EPSP in the dashed square of the middle panel is expanded below. The expansion shows a trace after application of sulpiride and CNQX. Each trace is an average from three consecutive traces. **e** IPSP amplitudes from each cell (circles) and means, for control recording and sulpiride application, during pre-drug period (pre) and 5 min after application (post). Lines connect data from the same recorded cells. **f** Percent pre-drug amplitude after application of control solution (gray bar) or sulpiride (pink bar). **e**, **f** n of recorded cells = 6 in each condition, n of recorded animals in **e** and **f** = 8 animals (six each for control and sulpiride; four pairs were from the same animals).

Optogenetic train stimulation (five pulses at 20 Hz) of EYFP-labeled DA neuron terminals elicited inhibitory postsynaptic responses (IPSPs) in 40 out of 61 recorded neurons. Among 40 cells, four cells were spontaneously active. In all of these four cells and two cells with induced firing by current injection, the firing was significantly reduced by the IPSPs (Supplementary Fig. 14b). The IPSPs were blocked by the D2 receptor antagonist sulpiride (Fig. 5d–f), suggesting that these sub-second IPSPs were mediated by G-protein coupled inward rectifying K$^+$ channels directly coupled with D2 receptors, as observed in the ventral midbrain DA neurons[31] and in striatal cholinergic interneurons[32]. Lastly we detected a very small EPSP that was CNQX sensitive, indicating glutamate co-transmission.

**VTA-to-LS DA signaling is necessary for normal aggression.** To investigate if DAergic input from the VTA into the LS is necessary to drive normal aggressive behavior we performed an optogenetic inhibition experiment, using archaerhodopsin, a light-gated proton pump which inhibits neural activity upon

photostimulation[33]. Specifically, we bilaterally injected male DAT$^{IREScre}$ mice with AAV5-EF1a-DIO-eArch3 or AAV5-EF1a-DIO-EYFP control virus into the VTA and implanted fiber optic ferrules above the MCLS (Fig. 6a–c and Supplementary Fig. 15a, b). Four weeks after viral injection and fiber implantation we tested mice in social isolation-induced aggression. Optogenetic stimulation with green light (532 nm, 10 mW, continuous 55 s) of DAergic fibers in MCLS in eArch injected animals during a fight bout reduced aggression (Fig. 6d). Overall, total fight time significantly decreased (Fig. 6e and Supplementary Fig. 16a–c). Inhibitory opsins can have excitatory side effects[34–37]. Such effects have largely been reported for extended opsin activation for more than 1 min, which is the reason why we had limited our activation protocol to 55 s. Furthermore, excitatory effects are mostly seen after but not during opsin activation (rebound effect and increased spontaneous release)[34,35]. Hence, in order to further minimize the risk that excitatory side effects might account for our behavioral effect, we limited the behavioral analysis to only the first aggression bout and to only the 55 s of light application. Impressively, we find close to a complete block

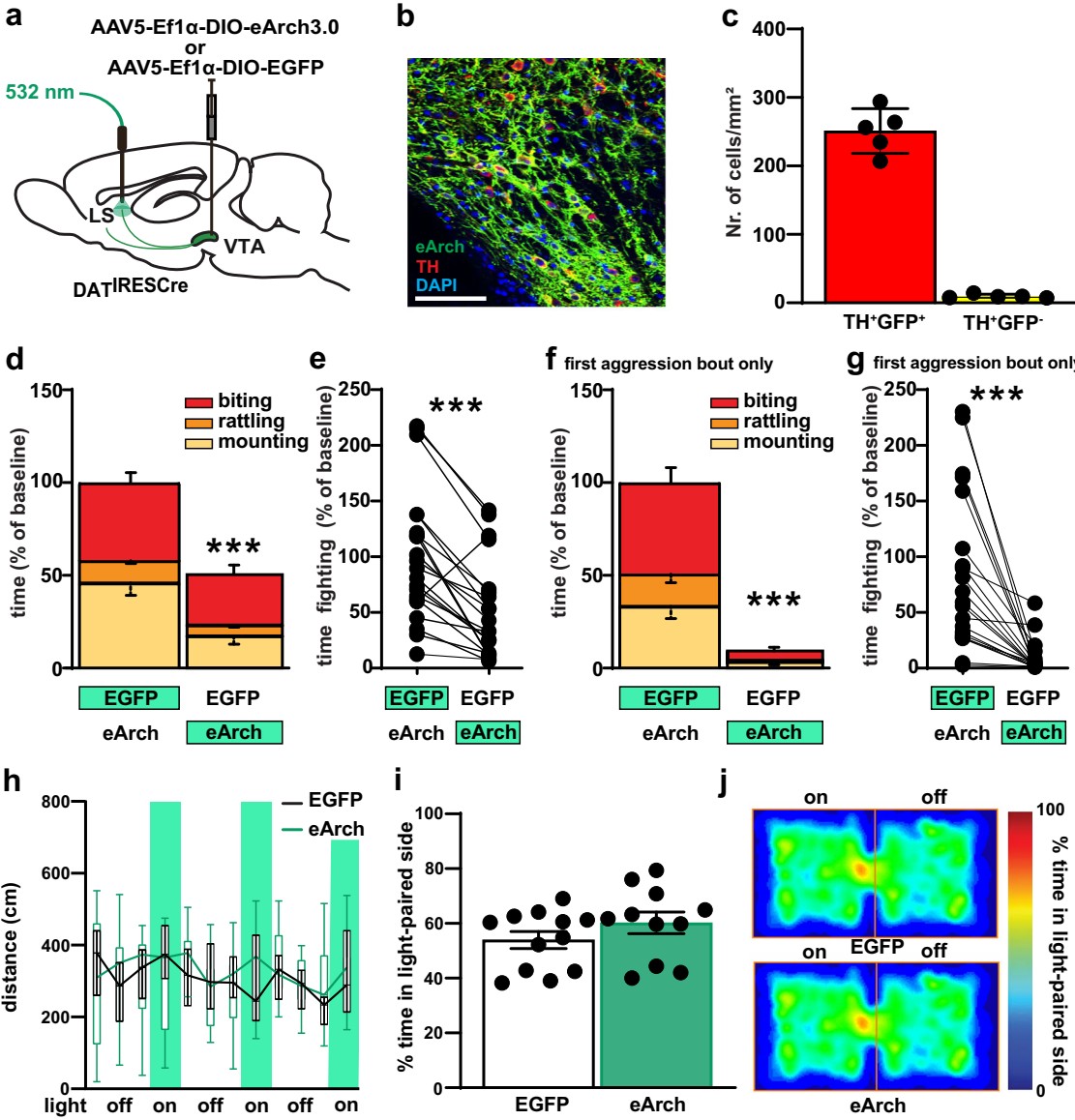

**Fig. 6 Optogenetic inhibition in VTA→LS DAergic terminals impair aggressive behavior. a** Schematic diagram for viral injection in DAT[IREScre] and terminal green light stimulation for optogenetic inhibition. **b** Representative image of cell type-specific eArch3.0-eYFP expression (green) in VTA DA neurons (red), with colocalized neurons in yellow, scale bar (100 μm). **c** Quantification of Arch expression in TH[+] neurons. **d, e** Inhibition (green) of LS DA terminals in DAT[IREScre] mice by optogenetic stimulation of eArch reduced aggression behavior. **f, g** Inhibition (green) of LS DA terminals in DAT[IREScre] mice by optogenetic stimulation of eArch abolished aggression behavior during the 55 s of light application triggered by the onset of the first aggression bout. **h** No locomotion activity change was detected after LS DAergic terminal inhibition in the open field, during alternating 3 min off and 55 s on (vertical green shading) bouts of continuous eArch stimulation. **i** During the RTPP task, the percentage of the time, over a 20 min session, spent in the stimulated zone was not significantly different between EYFP and eArch expressing mice. **j** representative heat maps of the time spent in the stimulated zone during RTPP. $P < 0.01$; ***$P < 0.001$ compared with their respective controls; mean ± SEM; $n = 21$ pairs, Box plots represent median and IQR and whiskers extend to maximum and minimum values (**h**). (Supplementary Data 1 for detailed statistics).

of aggressive behavior using this refined analysis (Fig. 6f, g). Locomotion in the open field test was not affected (Fig. 6h). Furthermore, optogenetic inhibition did not change behavior in the real-time place preference task (Fig. 6i, j). Together these data demonstrate that VTA-to-LS DAergic projections are necessary to establish normal aggression levels but play no role in normal locomotion or real time place preference.

**D2 signaling within the LS is necessary for VTA DAergic neurons to promote aggression.** TA projections to the LS might constitute the dominant pathway involved in DAergic promotion

of aggressive behavior. Alternatively, VTA DAergic projections to other brain regions might also contribute to DAergic facilitation of aggressive behavior. To evaluate the necessity of the LS pathway for VTA-triggered aggression, we infused sulpiride or vehicle into the LS (Supplementary Fig. 17), followed by VTA DAergic neuron activation during aggression testing (Fig. 7a). We found that D2 antagonism in the LS completely blocked VTA-triggered aggression (Fig. 7b, d–g), without altering general locomotor activity (Fig. 7c and Supplementary Fig. 18a–c). These data demonstrate that the VTA-to-LS DAergic pathway, via D2 activation on GABAergic LS neurons, is necessary to drive VTA-triggered aggression.

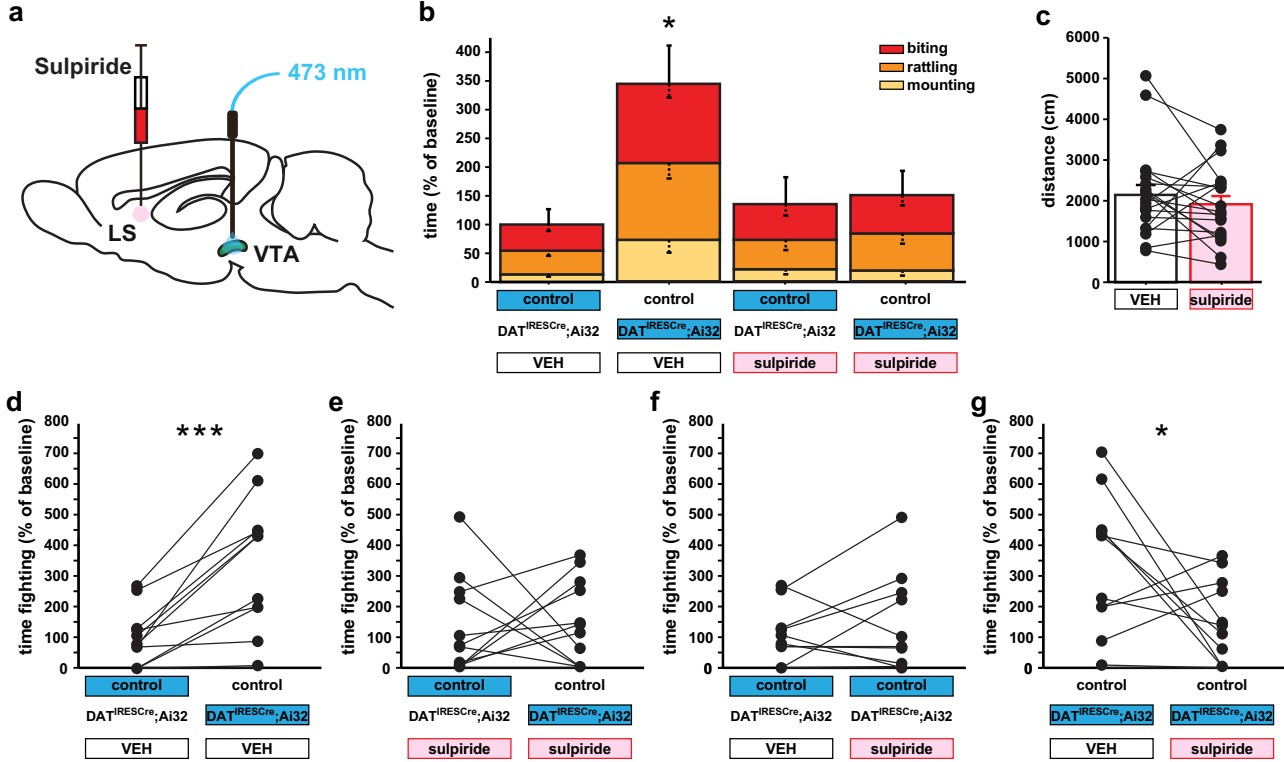

**Fig. 7 D2 receptor activation in the LS is necessary for VTA DAergic activity to increase aggression. a** The schematic diagram for local infusions of sulpiride in the LS followed by optogenetic stimulation (blue) of the VTA to evoke in vivo non-projection specific DA release. **b** Local sulpiride infusion into the LS prevents aggression elicited by optogenetic activation of VTA DAergic activity. **c** No difference in locomotion was detected between the groups following local infusion of sulpiride in the LS. Time-fighting, for individual pairs, is broken down by **d**, **e** treatment and **f**, **g** stimulation condition. *$P < 0.05$; ***$P < 0.001$ compared with their respective controls; mean ± SEM; $n = 11$ pairs (**b**, **d**–**g**) and 20/group (**c**). VEH, vehicle.

## Discussion

We identified a role for DAergic VTA-to-LS projecting neurons in aggression. We demonstrate that DAergic neurons densely innervate a subregion within the LS, which we termed MCLS (medial crescent lateral septum), where DA signaling is necessary and sufficient for orchestrating aggressive behavior.

The LS controls emotionality and aggression[38]. Several inputs into the LS have been identified that modulate these behavioral domains. For example, ventral hippocampus neurons projecting from CA3 to the LS play a major role in controlling anxiety[39] and contextual fear discrimination[40], while projections from CA2 to the LS regulate aggressive behavior[6,41]. Our findings add DAergic innervation from the VTA not just as a component that is sufficient to increase aggressive behavior but as a necessary input under baseline conditions to establish normal aggression levels. We find that terminal neurotransmitter release from DAergic axons in the LS inhibits GABAergic LS neurons via postsynaptic D2 receptor activation. A more detailed topographical analysis of PSC responses, as we have for example done in the striatum[32,42], was not possible here, because localized and not-abundant DA neuron inputs limited detection of PSCs in slice preparation. Thus, while we were able to qualitatively characterize identified PSCs, we cannot conclude on the precise quantitative aspect nor the topographical distribution for gradients in connectivity. Furthermore, how coordinated inputs from the VTA and other regions such as the hippocampus interact to control LS output and whether DA release in the LS also modulates inputs at the presynaptic levels remains to be investigated.

We find a dissociation in the behavioral roles of DAergic projections to the MCLS and NAc, where MCLS input drives aggression while NAc input drives reward and motor activity.

This finding is in line with the functional organization of monoamine systems, where relatively few monoaminergic neurons reside as densely packed clusters in the midbrain and brainstem but project widely to distributed target areas, modulating behaviors that are controlled by multiple target regions. The result of manipulating DAergic activity is thus greatly dependent on which specific DAergic target site is being manipulated. Our data indicate that separate DAergic pathways drive distinct behavioral components of aggression. While NAc DA does not initiate aggression, it might ascribe positive valence to LS-triggered fighting behavior and thereby contribute to proactive aggression.

Mirroring the dissociation at the behavioral level, we also find a dissociation at the anatomical level. Specifically, we identified VTA DAergic neurons that innervate the MCLS as a unique neuronal population, that is different from DAergic neurons that project to the NAc. Cell bodies reside at distinct locations within the VTA and do not co-innervate each other's target region. DAergic fiber distribution within the LS is in line with TH immunoreactivity in the LS of rats and mice[25,43]. Neurons projecting to the LS are concentrated in a relatively more anterior and medial location (Supplementary Movie 1) within the VTA, than those projecting to the NAc.

The absence of immunohistochemical evidence of collateralization comports with each target site controlling distinct functional outputs[44]. Future studies of genetic and molecular markers in MCLS-projecting and NAc-projecting cell populations might help to better understand their unique properties and their ontogeny. Given the known association between NAc DA release and aggression[20] as well as the necessary role for D2 expressing NAc neurons in the rewarding properties of aggression[45], it will

be important to understand the coordinated activities of both cell populations during behavior. Using fiber photometry to monitor $Ca^{2+}$ transients with genetically encoded fluorescent $Ca^{2+}$ sensors in projection specific DAergic neuronal populations would for example reveal their real time activity by proxy of CA2+ transients[46,47]. Alternatively, using fiber photometry to monitor DA release with genetically encoded fluorescent DA sensors in DAergic projection target regions would reveal their real time activity by proxy of extracellular DA content[48]. Using these techniques to test whether and how the LS-projecting and NAc-projecting VTA DA neurons change their activity pattern during the natural aggression behavior will be important to better understand the coding properties of both pathways. While we hypothesize that the VTA(DA)-to-LS pathway is important for the initiation and maintenance of fighting behavior while the VTA(DA)-to-NAc pathway assigns valence and establishes reinforcing properties to aggressive behavior, this hypothesis can only be properly scrutinized by analyzing neuronal activities during behavior. Future work will directly assess such temporal interplay between spatially segregated DAergic pathways in complex and dynamic behavior.

Our identification of a population-specific and projection-specific DAergic input to the LS that exerts a powerful influence over aggressive behavior effectively join two major streams of aggression research—the hyper-DAergic model of aggression[12,49] and the classic LS-hypothalamic aggression axis involving "septal rage"[5,6] and the "hypothalamic attack area"[7]. Psychological and behavioral data furthermore support the classification of aggression into two dominant forms: (1) reactive aggression which occurs impulsively in response to perceived external threat and (2) proactive aggression that is premeditated and directly motivated by a drive for appetitive reward[50]. VTA DAergic activity is prominently involved in reward coding[51]. Hence, one might expect a role for VTA-to-LS input in proactive aggression[50,52,53]. However, our findings of DAergic LS input promoting aggression without affecting real time place preference indicate a role for DA in reactive aggression[54]. Extreme forms of impulsive aggression are directed against the opposite sex[45]. Indeed, we find that male mice also attack females when the VTA-to-LS input is stimulated. Aggressive, but not sexual behavior towards females[55,56], has been discussed as displaced aggression towards an inappropriate target[57]. This description is reminiscent of an impulsive form of aggression seen in humans with intermittent explosive disorder, who display inappropriate and violent outbursts towards targets that don't typically elicit aggression[50,58]. The lack of reward coding together with indiscriminate targeting further implicates the VTA-LS hyper-DAergic aggression circuit in impulsive and reactive aggression, rather than outcome-oriented and proactive aggression.

Our findings suggest that genetic and environmental factors that increase LS DAergic activity can be risk factors for aggression-related psychopathology. Acute and repeated exposure to DA-boosting drugs such as psychostimulants can increase aggression[59]. In mice, transient increase of DA signaling during adolescence can even permanently increase aggression throughout life[22]. Also, genetic variants, that impact DAergic activity, can increase baseline aggression and thereby contribute to clinical psychopathy. For example, the low-activity *COMT* (V158M) allele is associated with aggressive personality traits[60], and the 10-repeat allele of *DAT1* is linked to aggressive and antisocial dispositions[61]. Our data indicate that such genetic risk might act via increasing (re-) activity of the DAergic VTA-LS pathway[12,22]. Furthermore, our findings indicate that clinically established pharmacotherapy, currently used to manage pathological aggression, acts on D2 receptors within the LS to specifically reduce aggression, and not via DA receptor-mediated sedation[19,62]. This insight underscores the probable utility of

selectively targeting broadly expressed GPCRs on specific cell types[63]. Together our data reveal DA circuit mechanisms underlying aggressive behavior in mice with potentially translational relevance for understanding normative and pathological aggression in humans.

## Methods

**Animals.** All animal testing was conducted under protocols approved by Columbia University and New York State Psychiatric Institute Institutional Animal Care and Use Committees. Adult mice aged 12–32 weeks were used for all studies. Mice were housed at 18–23 °C, with 40–60% humidity under a 12 h light-dark cycle (lights on at 7 am), with food and water available ad libitum. DAT[IRESCre23] and Ai32 (RCL-ChR2(H134R)/EYFP)[24] mice were crossed to produce experimental cohorts consisting of DAT[IRESCre]; Ai32 and single mutant controls on a mixed F1 background (C57BL/6J × 129S2/129SvEv/Tac). DAT[IRESCre]; Ai32 mice expressed ChR2 in > 95% of DAergic neurons, based on immunohistochemistry[64]. Thus, this strategy allowed us to interrogate almost all of DAergic inputs in target regions without variability. For the RNAscope experiment, Drd2-EGFP, GENSAT-S118Gsat/Mmnc mice were used. Given that female mice do not display territorial aggression, only male mice were investigated.

**Viruses.** AAV5-ef1α-DIO-hChR2(H134R)-EYFP, AAV5-EF1a-DIO-eArch3.0-EYFP, and AAV5-EF1a-DIO-EYFP viruses were purchased from the University of North Carolina vector core. hEF1α-LS1L-GCaMP6s purchased from the Massachusetts General Hospital Neuroscience Center vector core. All AAV titers ranged from $3.00 \times 10^{12}$ to $4.30 \times 10^{13}$ genomic copies per ml. HSV titer was $1.00 \times 10^{9}$ genomic copies per ml.

**Stereotaxic surgery and viral vectors.** All behavioral mice were implanted with optical fiber ferrules (Ceramic ferrules (OD 1.2 mm): Precision fiber products; Optical fiber (OD 0.2 mm): Thorlabs, Newton, NJ, USA). During surgery, mice were anesthetized with ketamine (10 mg/kg body weight) and xylazine (50 mg/kg) or isoflurane (1–1.5%) solution and placed in a stereotactic frame.

For genetic approach optogenetic stimulation experiments, the optical fiber ferrules were implanted over the VTA (AP: −3.5; ML: ±0.5; DV: −4.2), the SNc (AP: −3.5; ML: ±1.6; DV: −4.4), the LS (AP: 0.6; ML: ±0.4; DV: −2.8) and the NAc (AP: 1.6; ML: ±1.4; DV: −3.8).

For the optogenetic activation experiment of VTA-to-LS, 500 nl of AAV5-ef1α-DIO-hChR2(H134R)-EYFP or AAV5-EF1a-DIO-EYFP was injected ipsilaterally into VTA and optical fiber ferrule was implanted over LS in the same hemisphere.

For the optogenetic inhibition experiment of VTA[LS], 500 nl of AAV5-EF1a-DIO-eArch3.0 or AAV5-EF1a-DIO-EYFP was injected bilaterally into VTA and optical fiber ferrule was implanted over LS in both hemispheres.

For the retrograde labeling experiment, 200 nl of hEF1-LS1L-GCaMP6s was injected ipsilaterally into LS and NAc, respectively. Brains were collected 4 weeks later.

For the local infusion experiments, DAT[IRESCre]; Ai32 mice were unilaterally implanted with a guide cannula (26 gauge; PlasticOne) directly above the LS (AP: +0.6 mm, ML: −0.4 mm, DV: −2.3 mm).

**Behavioral testing.** To assess aggressive behavior, we used the isolation-induced aggression paradigm[65]. The home cage was divided in half by a perforated partition made of clear plastic, pairing DAT[IRESCre]; Ai32 mutant mice with controls. Mice were housed for 10 days before the experiment was performed. For the optogenetic activation experiment, on test day, dividers were taken out and blue light pulses (473 nm, 20 Hz, 10 ms pulse duration) were delivered to the region being investigated. Only one mouse of the pair was stimulated during each 15 min encounter. The time spent fighting was scored as a sum of the time spent biting, tail rattling, and mounting. For the optogenetic inhibition experiment, on the test day, whenever the fighting bout started, one mouse of the pair was stimulated with continuous green light (532 nm, 10 mW) for 55 s in the lateral septum.

The assay was moderately adapted to investigate the effect of local infusions of D2 receptor antagonist, sulpiride, in the LS. Fifteen minutes before the test, mice received 0.15 µl volume infusions of either sulpiride (Sigma Aldrich; 0.05 µg/mouse) or vehicle (0.9% physiological saline) microinjected at a rate of 0.05 µl/min over 3 min. The injection cannula (28 gauge; PlasticOne) was left in place for 2 min after the infusion and the mice were returned to their home cages before aggression testing using the stimulation protocol described above.

Locomotor activity in an open field, in response to optogenetic activation, was assessed in Plexiglas activity chambers as previously described[66]. The open field is a standard test for locomotor behavior. It consists of a simple square enclosure that is equipped with infrared detectors to track animal movement in the horizontal and vertical planes. Measures of total distance covered during locomotion are used as an index of activity. Mice were placed into the center of the open field and activity was recorded for 12 min in 3 min bouts of alternating light-off and light-on conditions. Testing took place under bright ambient light conditions and the total distance traveled was measured.

Reward-based preference, as a function of stimulation, was evaluated in the real-time place preference test. Mice implanted in the NAc and the LS were placed in a custom-made behavioral arena (18″ × 10″ × 8″) for 20 min. The arena was divided into two chambers, each with different visual and tactile cues. Counterbalancing for genotype and stimulation side, each mouse was placed in the non-stimulated side at the onset of the experiment and delivered a 20 Hz laser stimulation (stimulation protocol above) each time the mouse entered the assigned stimulation side. The stimulation was terminated the second the mouse crossed back into the non-stimulation side. Time spent in each chamber (stimulated vs. non-stimulated) was recorded using the ANY-maze video tracking software (Stoelting Co.).

**Histology.** Animals were deeply anesthetized with ketamine/xylazine and perfused with phosphate-buffered saline (PBS) followed by 4% paraformaldehyde (Electron microscopy sciences, Cat# 15714-S) in PBS. Brains were then removed and post-fixed overnight (4% paraformaldehyde in PBS), and subsequently cryoprotected in 30% sucrose and frozen. Coronal sections (50 μm thick) were cut either with a cryostat (Leica Biosystems) or vibratome (Leica Biosystems) and fluorescence immunohistochemistry was performed on free-floating sections.

**Immunohistochemistry.** Sections were first rinsed with PBS and then blocked with buffer (PBS, 5% normal goat serum (NGS), 5% normal donkey serum, and 0.5% Triton X-100). This was followed by overnight incubation with primary antibody and eventually conjugated to a fluorophore with the appropriate secondary antibody. The sections were incubated in primary antibody for 24 h at room temperature, washed 3× with PBS, and then incubated with secondary antibody for 1 h at room temperature. After washing 3× with PBS the sections were incubated with DAPI (1:20,000; catalog no. D1306, Invitrogen) and transferred into PBS solution. Finally, the sections mounted on Superfrost slides (Fisher Scientific; catalog no.12-550-15) and cover slipped with Prolong Gold Antifade (catalog no. P36930; Invitrogen).

For optic fiber location identification (Supplementary Figs. 3a–d, 5, 8, and 17) GFP, immunohistochemistry was performed using a rabbit primary antibody against GFP diluted in blocking solution 1:1000; Life Technologies, Grand Island, NY, USA. As a secondary antibody we used Cy3 donkey anti-rabbit, diluted in blocking solution 1:350; Jackson Immunoresearch, West Grove, PA, USA.

For TH and GFP double staining, immunohistochemistry was performed using a rabbit anti-TH primary antibody (1:1000; catalog no. AB152, Millipore Sigma, Darmstadt, Germany) and a chicken anti-GFP primary antibody (1:5000; catalog no. ab13970; Abcam, Cambridge, UK). As secondary antibody, we have used donkey anti-rabbit conjugated to Alexa 647 (1:500; catalog no. ab150075; Abcam, Cambridge, UK) and goat anti-chicken conjugated to Alexa 488 (1:500; catalog no. 103-545-155; Jackson ImmunoResearch, West Grove, PA, USA).

**Projection analysis.** For projection analysis, four brains were included from each of the injection sites, NAc, and LS. After the TH and GFP double staining, a custom-build automated slide scanner AZ100 microscope equipped with a 4 × 0.4 NA Plan Apo objective (Nikon Instruments Inc.) and P200 slide loader (Prior Scientific), controlled by NIS-Elements using custom acquisition scripts (Nikon Instruments Inc.) was utilized. This imaging provided automated high-throughput brain-wide imaging of the whole brain. Images were processed and reconstructed with an Image J plugin called Brain J[28].

For consistency between brains, only sections registered in the range +2.7 to −4.2 AP from Bregma were selected, and the same number of images were analyzed for each ROI. Non-rigid free form registration of coronal mouse brain sections was done in RStudio with the open-source *WholeBrain Software*[27]. Briefly, each image was first matched to a correspondent plate of the Mouse Allen Brain Reference Atlas CCF v3 from Allen Institute Common Coordinate Framework[67]. Because mouse brains were cut at 50 μm and the reference Atlas spacing between plates is 100 μm, two sections were registered to the same correspondent plate. The contour of the brain section was segmented out using the autofluorescence of the down-sampled section itself, and a set of correspondence points that align the selected reference atlas plate with the tissue-section was automatically generated by the "thin-plate splines algorithm". Landmarks such as ventricles were used to manually adjust add or remove correspondent points when necessary. The two sets of reference points (atlas and tissue-section reference points) constitute a 2D final mapping of each brain region along the anteroposterior axis.

To measure the extension of projections in each region of interest (ROI), all tissue images containing the ROI were first selected based on the registration mapping for each brain and ROI contour coordinates extracted by a custom R-script. The coordinates were inputted to a custom Image-J macro to draw the ROI contour in the correspondent 16-bit image, subtracting the background from the images (Process\Subtract Background), and a rolling ball radius of pixels, was chosen to improve the contrast. The macro continued in selecting the contour where the threshold tool was applied to segment the projections based on the intensity of pixels. For each brain, the total projection in the ROI was calculated as the sum of the segmented pixel count in each image selected for that ROI. The pixel count in each ROI was normalized by the pixel count in the brain area representing the injection site. The density was calculated by dividing the pixel count of the segmented projection by the total volume in the ROI expressed in the pixel. Segmentation of somas of GFP+ neurons in VTA and PAG was done on all sections containing those brain regions by multiresolution decomposition on the *WholeBrain Software*[27]. For one brain for each injection type, the segmented neurons were projected into a 3D reconstruction of the individual brain sections by using the "glassbrain" function of the software[27], and color-coded based on the location of the injection (magenta for LS and green for NAc). The 3D brain render with the segmented somas from each mouse was rotated on the *y*-axis in RStudio to generate a single animation (Supplementary Movie 1). A 3D render of VTA and PAG was also created by using the cocoframer package (https://github.com/AllenInstitute/cocoframer) and superimposed frame by frame to the animation video generated (gray area after one full rotation rotation).

Moreover, for the preparation of Fig. 2k from each animal, a binarized stack of ten images was created by selecting ten of the 16-bit tissue images containing VTA (range −2.8/−3.8 AP). Neurons were segmented by applying a erode/dilate/open filter on the stack, and the images superimposed by max projection. The LUT of the resulted image was modified in magenta (LS) or green (NAc) and finally the images for each animal were superimposed to one representative DAPI section and the VTA outlines drawn with white dashed lines.

**Fluorescent in situ hybridization with RNAscope®.** The expression and colocalization of dopamine receptor 2 (Drd2), glutamate decarboxylase (GAD1), in mutant mice were verified using an in-situ hybridization assay. Drd2-EGFP mice were culled by cervical dislocation and the brains removed from the skull and snap-frozen on powdered dry ice. Fresh frozen 16 μm sections were cut coronally through the lateral septum area using a cryostat (CM3050S Leica Biosystem) and thaw-mounted onto Superfrost Plus Microscope Slides (Fisher Scientific; catalog number 12-550-15). The samples were prepared and pretreated for the assay using the RNAscope® Sample Preparation and Pretreatment Guide for Fresh Frozen Tissue (Advanced Cell Diagnostic; document number 320513) with the following adjustments. The slides were fixed in 4% PFA for 45 min at 4 °C before being dehydrated at room temperature using graded ethanol (50, 70, 100, and 100%) and then air-dried. To increase the adhesiveness of the sections a baking step of the slides was performed for 15 min and 60 °C. At last, the sections were pretreated with protease IV for 30 min. The probes for D2 andGAD1 provided by Advanced Cell Diagnostics (catalog numbers: D2 #406501, GAD1 #400951) were conjugated to Atto 550, Atto 647 respectively. The procedure for in situ detection was performed using RNAscope® Fluorescent Multiplex Reagent Kit (Advanced Cell Diagnostics, document number 320293) according to the manufacturer's instructions. The HybEZ™ oven (Advanced Cell Diagnostics, PN 321710/321720) was used in the heating steps and the slides were mounted using Prolong Gold Antifade (Thermo Fisher Scientific; catalog no. P36930). Images were acquired using a laser confocal scanning microscope (Leica, TCS SP8) and positive labeling was counted using LASx software (Leica, TCS SP8).

**Slice electrophysiology.** DAT$^{IRESCre}$;Ai32 mutant mice (P72-93) were anesthetized with a ketamine (90 mg/kg)/xylazine (7 mg/kg) mixture. After confirmation of full anesthesia, mice were decapitated and brains quickly removed into ice-cold high glucose artificial cerebrospinal fluid (ACSF) (in mM: 75 NaCl, 2.5 KCl, 26 NaHCO$_3$, 1.25 NaH$_2$PO$_4$, 0.7 CaCl$_2$, 2 MgCl$_2$ and 100 glucose, pH 7.4) saturated with carbogen (95% O$_2$ and 5% CO$_2$). Coronal sections of the septum (300 μm, 0.1 mm anterior to bregma) were made with a vibrating microtome (VT1200S, Leica). Sections were incubated in high glucose ACSF at room temperature for at least 1 h for recovery, then a slice was transferred to the recording chamber (submerged, 500 μl volume) on the stage of an upright microscope (BX61WI, Olympus), continuously perfused with standard ACSF (in mM: 125 NaCl, 2.5 KCl, 25 NaHCO$_3$, 1.25 NaH$_2$PO$_4$, 2 CaCl$_2$, 1 MgCl$_2$, and 25 glucose, pH 7.4) saturated with carbogen. Neurons were visualized using enhanced visible light differential interference contrast (DIC) optics with a scientific c-MOS camera (ORCA-Flash4.0LT, Hamamatsu photonics). Recording patch pipettes were fabricated from standard-wall borosilicate glass capillary with filament (World Precision Instruments). Pipette resistance was 3–7 MΩ. Composition of intracellular solution for firing and the slow EPSC recording was (in mM): 135 K+-methane sulfonate (MeSO4), 5 KCl, 2 MgCl$_2$, 0.1 CaCl$_2$, 10 HEPES, 1 EGTA, 2 ATP and 0.1 GTP, pH 7.25. Fast current-clamp recording was performed with an Axopatch 200B amplifier (Molecular Devices). Liquid junction potential (~10 mV) was adjusted online. Series resistance was 15–27 MΩ and was not compensated. The recording was performed in the LS just dorsal to the fornix, in areas with high EYFP fluorescence of DA neuron terminals, from large pyramidal-shaped/triangular-shaped neurons located. One to two slices per animal included the area innervated by DA neurons, and only one slice includes sufficient amount of DA fiber innervation to evoke synaptic responses. In some animals, neither of the slices had sufficiently dense DA fiber innervation, and those slices were discarded. Synaptic responses were evoked by 5 ms duration field illumination with a high-power blue LED (Thorlabs) delivered in trains of five pulses at 20 Hz, repeated at 1 min intervals. For pharmacological studies, drugs were delivered by perfusion. Recordings were done at 31–33 °C (TC 344B Temperature Controller, Warner Instruments). Recorded data were digitized at 5 kHz (Digidata 1550A, Molecular Devices), filtered at 5 kHz with a 4-pole Bessel filter, and recorded using pClamp 10 (Molecular Devices). Data were analyzed with Axograph X (Axograph Scientific).

Average firing frequencies were calculated in 0.5 s windows: for baseline, 0.5–1 s prior to the onset of a training stimulus, and for stimulus effects, 0.1–0.6 s after the onset of the train. PSC peak amplitude was measured in averages of three consecutive traces.

**Statistical analysis**. Statistical analysis for behavior was performed using StatView 5.0 software (SAS Institute, Cary, NC, USA) or Prism9 software (GraphPad) and Microsoft Excel. Data were analyzed using Student's *t*-test, one-way or two-way ANOVA with Student–Newman–Keuls post hoc testing as indicated. The criterion for significance for all analyses was *$P < 0.05$; **$P < 0.01$; ***$P < 0.001$. For the physiology, statistical tests were done using SPSS 23 (IBM). Since sample sizes were small, for the independent sample *t*-test, we did not assume equal variance. Sample size estimation was done by G*Power 3.1, with $P = 0.05$, power = 0.9. The effect size was estimated from previous experiments. Results are expressed as mean ± SEM.

**Reporting summary**. Further information on research design is available in the Nature Research Reporting Summary linked to this article.

## Code availability

Associated Code can be found here: https://github.com/albilard/Neuro_quantify_projection.

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

## Acknowledgements

We would like to thank Zuckerman Institute's Cellular Imaging platform for instrument use and technical advice. This work has been supported by the National Institute of Mental Health (by R01 MH099118 to M.S.A., R01MH111918 to D.D.), the National Institute on Drug Abuse (R01 DA038966 to S.R.) and the Sackler Institute for Developmental Psychobiology (M.S.A.).

## Author contributions

D.M. and R.S. performed all behavioral, anatomical, and confocal microscopy experiments, N.C. performed electrophysiological experiments, A.Z. assisted with behavior, histology, RNAscope experiment, A.A.M. helped in imaging experiments, A.M. wrote custom image analysis scripts and conducted the quantification of brain-wide anatomical data, D.D. supervised the anatomical quantification S.R. supervised electrophysiological experiments, M.S.A. supervised the study. D.M., R.S., N.C., S.R., and M.S.A. wrote the manuscript. Both D.M. and R.S. contributed equally and have the right to list their name first in their CV. All authors contributed to the article and approved the submitted version.

## Competing interests

The authors declare no competing interests.

## Additional information

 **Peer review information** *Nature Communications* thanks Minmin LUO and the other, anonymous, reviewer(s) for their contribution to the peer review of this work. Peer reviewer reports are available.

