## [Peer Review File · Nature Communications]

Dopamine promotes aggression in mice via ventral tegmental area to lateral septum projectionsEditorial Note: This manuscript has been previously reviewed at another journal that is not operating a transparent peer review scheme. This document only contains reviewer comments and rebuttal letters for versions considered at *Nature Communications*.

REVIEWER COMMENTS

Reviewer #1 (Remarks to the Author):

Aggression and violence present a critical public health concern. Previous reports showed that neurons in the lateral septum (LS) and dopamine (DA) neurons in the ventral tegmental area (VTA) are involved in aggression in humans and animals. The same group of this paper has reported that optogenetically activating VTA DA neurons increases aggression (Yu et al. *Mol Psychiatry*. 2014). Here, Mahadevia et al. showed that the DAergic pathway from the VTA to the LS, but not the VTA-NAc pathway, is critical for promoting aggression. Interestingly, VTA DA neurons projecting to the LS and those to the NAc represent distinct neuronal populations. The LS-projecting VTA DA neurons act on D2 receptors to inhibit GABAergic neurons in the LS, thus promoting aggression. These findings have some important implications in the structural and functional specialization in the midbrain dopamine subsystems and in the circuit mechanisms underlying aggression.

The authors have carried out a number of experiments to address most of my previous concerns. The manuscript has been substantially improved. I have only one remaining concern that the authors may want to address experimentally or via discussion, at the discretion of the Editor. Specifically, the authors have performed optogenetic activation, and in the revision, inhibition, to examine the role of the VTA-LS projection in aggression. It would be ideal to test whether and how the LS-projecting VTA DA neurons change their activity pattern during the natural aggression behavior. Is the activity of these DA cells important for initiating aggression or for reinforcing aggression-related behavior? This can be tested by fiber photometry using Ca²⁺ sensors and/or genetically-encoded fluorescent DA sensor.

Reviewer #2 (Remarks to the Author):

The authors have addressed several of my concerns and I find the manuscript significantly improved by the addition of new experiments. The independence of the VTA-LS and VTA-NAc pathways is now better substantiated and quantified, as is the colocalization of drd2 and GAD1. The new optogenetic experiments also further support the conclusion for a specialized role in aggression for the VTA-LS pathway. The concerns about how much the field is advanced by the study remain - the work presents a refinement of the authors previous work on the role VTA in aggression and narrows down the neural circuit - but I will leave this up to the editors.

I have a few remaining technical concerns that should be addressed:

1 - The new experiment using terminal inactivation with eArch3 is a great addition to demonstrate a role for the VTA-LS pathways in natural aggression. The authors observe an effect congruent with their hypothesis and inhibition of VTA-LS terminals, but there have been several papers demonstrating that direct terminal illumination of both ion pump and ion channel based optogenetic tools can have excitatory side effects (Mahn et al. 2016 *Nat Neuroscience*; Raimondo et al. 2012 *Nat Neuroscience*; Malyshev et al. 2017 *Neurosci Letters*; reviewed in Wiegert et al. 2017 *Neuron*). It would be important to exclude that this is happening, which could be done in slice recordings (eg: optogenetic stimulation of VTA-LS terminals expressing eArch3 while recording from downstream neurons)

2 - I still have concerns regarding the electrophysiology experiments:

a - In response to my previous comments, the authors state that there were 12 cells that fired spontaneously, 8 of which were not inhibited by stimulation of the VTA-LS pathway, and were therefore excluded (2 additional cells from current injection were added, giving the total of 6 datapoints shown in the figure). I must be missing something here. As it is stated, it sounds like the authors did an experiment to test for an effect using a group of cells, and then excluded from the analysis the cells that did not show the effect they wanted. This is of course completely unacceptable from a statistical point of view. If 8/12 cells did not show an effect, then clearly there

is no effect on firing rate during VTA-LS stimulation. This is a very significant concern.

b - I take the authors argument that the IPSPs are likely not GABAergic, the sulphiride data are convincing. However, related to the point above, it is important to see the distribution of IPSP magnitudes elicited by VTA-LS stimulation. The authors state that 40 cells showed a response, I would like to see the distribution for these responses and statistics showing that there is significant inhibition of membrane potential across the entire population of cells that had a response. Also, when describing this experiment the authors state that 65% of cells show a response, $n=40$. I believe that this should be $n=62$ (with $40 = 62 \cdot 0.65$).

c - there are still some statements that are not backed up by appropriate quantification: "The size of the IPSPs was affected by how DA neuron projections were included in recorded slices". Where is this shown? Also, "in most cases depolarizing current injection caused burst firing", where is the data for this, other than the example shown?

3 - Please include the number of animals for the RNAscope experiments, and show the variance of the quantification across animals.

Reviewer #3 (Remarks to the Author):

The authors have done a nice job in revising this manuscript. I have no further questions or comments.

REVIEWER COMMENTS

Reviewer #1 (Remarks to the Author):

Aggression and violence present a critical public health concern. Previous reports showed that neurons in the lateral septum (LS) and dopamine (DA) neurons in the ventral tegmental area (VTA) are involved in aggression in humans and animals. The same group of this paper has reported that optogenetically activating VTA DA neurons increases aggression (Yu et al. Mol Psychiatry. 2014). Here, Mahadevia et al. showed that the DAergic pathway from the VTA to the LS, but not the VTA-NAc pathway, is critical for promoting aggression. Interestingly, VTA DA neurons projecting to the LS and those to the NAc represent distinct neuronal populations. The LS-projecting VTA DA neurons act on D2 receptors to inhibit GABAergic neurons in the LS, thus promoting aggression. These findings have some important implications in the structural and functional specialization in the midbrain dopamine subsystems and in the circuit mechanisms underlying aggression.

The authors have carried out a number of experiments to address most of my previous concerns. The manuscript has been substantially improved. I have only one remaining concern that the authors may want to address experimentally or via discussion, at the discretion of the Editor. Specifically, the authors have performed optogenetic activation, and in the revision, inhibition, to examine the role of the VTA-LS projection in aggression. It would be ideal to test whether and how the LS-projecting VTA DA neurons change their activity pattern during the natural aggression behavior. Is the activity of these DA cells important for initiating aggression or for reinforcing aggression-related behavior? This can be tested by fiber photometry using Ca²⁺ sensors and/or genetically-encoded fluorescent DA sensor.

→ We are happy to see that Reviewer #1 finds our revisions to have substantially improved the manuscript. We agree that the study of pathway specific activity during behavior is an important follow-up to the current story, as it will yield insight into how the dopamine system relays aggression-relevant information to its target regions. We hypothesize that VTA DAergic neurons projecting to the LS and to the NAc are active during aggression, and that the VTA-to-LS pathway leads to the initiation and maintenance of aggressive behavior (as we have shown here), while the VTA-to-NAc pathway assigns positive valence to aggressive behavior. We now elaborate on these aspects in the discussion section.

Reviewer #2 (Remarks to the Author):

The authors have addressed several of my concerns and I find the manuscript significantly improved by the addition of new experiments. The independence of the VTA-LS and VTA-NAc pathways is now better substantiated and quantified, as is the colocalization of drd2 and GAD1. The new optogenetic experiments also further support the conclusion for a specialized role in aggression for the VTA-LS pathway. The concerns about how much the field is advanced by the study remain - the work presents a refinement of the authors previous work on the role VTA in aggression and narrows down the neural circuit - but I will leave this up to the editors.

I have a few remaining technical concerns that should be addressed:

1 - The new experiment using terminal inactivation with eArch3 is a great addition to demonstrate a role for the VTA-LS pathways in natural aggression. The authors observe an effect congruent with their hypothesis and inhibition of VTA-LS terminals, but there have been

several papers demonstrating that direct terminal illumination of both ion pump and ion channel based optogenetic tools can have excitatory side effects (Mahn et al. 2016 Nat Neuroscience; Raimondo et al. 2012 Nat Neuroscience; Malyshev et al. 2017 Neurosci Letters; reviewed in Wiegert et al. 2017 Neuron). It would be important to exclude that this is happening, which could be done in slice recordings (eg: optogenetic stimulation of VTA-LS terminals expressing eArch3 while recording from downstream neurons)

→ We thank the reviewer for this cautionary comment. We are aware that the use of inhibitory opsins for terminal inhibition has its limitations. However, we think in our case the limitations do not affect our experiment and data interpretation. Specifically, Mahn et al. (2016) report that sustained (> 1 min) activation of Arch increases the frequency of spontaneous transmitter release. We had designed our experiments to avoid this possible confound by limiting Arch activation to 55 seconds. Furthermore, the limitations of terminal NpHR activation described for example by Raimondo et al. (2012) concern the activity of the synapse 'after' silencing with NpHR, when excitability is increased by changes in the Cl⁻ reversal potential, but not 'during' silencing. This increase of spontaneous release is also discussed in detail by Wiegert et al. (2017). To address this possible confound of increased spontaneous release after terminal inhibition, we re-analyzed our behavioral data, only looking at the 55 seconds of stimulation following the first aggression bout in each pair. These new data are presented in new Figure 6. We basically find a complete inhibition of aggressive behavior during light administration in DAT-Cre eArch mice. This finding rules out that post-light rebound effects or increased spontaneous activity might confound our data interpretation.

The suggestion of slice recordings to exclude that excitatory side effects could confound our data interpretation is at first sight reasonable. However, we would not expect a comparable excitatory effect size following terminal ChR2 and Arch activation. Furthermore, given that we record not direct glutamatergic or GABAergic activity, but a modulatory inhibitory effect of dopamine in a region of the brain that is sparsely innervated by dopaminergic fibers (when compared to the striatum for example), this proposed “simple” experiment is actually not feasible. To exclude a false negative result for an effect size of for example 20% the size of the ChR2 triggered effect size, we would need to record from at least 5 times as many neurons, a task that we respectfully deem unfeasible. Simply put, although recording from the LS itself in slices may not be difficult, recording from this particular location of the LS receiving DA neuron inputs is extremely difficult (low yield) and this difficulty inevitably limits the range of capable experiments, such as detecting a relatively small excitatory side effect.

2 - I still have concerns regarding the electrophysiology experiments:

a - In response to my previous comments, the authors state that there were 12 cells that fired spontaneously, 8 of which were not inhibited by stimulation of the VTA-LS pathway, and where

therefore excluded (2 additional cells from current injection were added, giving the total of 6 datapoints shown in the figure). I must be missing something here. As it is stated, it sounds like the authors did an experiment to test for an effect using a group of cells, and then excluded from the analysis the cells that did not show the effect they wanted. This is of course completely unacceptable from a statistical point of view. If 8/12 cells did not show an effect, then clearly there is no effect on firing rate during VTA-LS stimulation. This is a very significant concern.

→ Because of very localized and not-abundant inputs, recording of DA neuron synaptic responses in the LS is not trivial. The LS location recipient to DA neuron inputs is very narrow, V-shape and canted with respect to the coronal plane. This narrow 'V' must be included in the LS slice and accessible from the surface of the slice for successful recording. Therefore, unlike other locations (e.g. the striatum), 'false negatives' due to slicing are presumably frequent in this particular brain slice preparation, and we cannot discern whether lack of synaptic response is due to lack of real synaptic connections or artifacts introduced by slicing, e.g. truncated dendrites of the recorded cell. While we report the "success rate", the main purpose of this slice study was to characterize identified responses via well-controlled drug application experiments. We detected 41 inhibitory responses in 60 patched cells. For the non-responding cells we cannot conclude if they are false negatives due to the slice preparation or if they actually do not receive DA input in vivo. To understand the nature of the identified inhibitory input we selected cells with inhibitory responses and studied those further. To analyze the effect of elicited IPSPs on firing frequency, we needed cells that showed an IPSP and fired. Only 4 out of 12 spontaneously active cells fit this criterion (two additional cells were injected with current to elicit firing). We cannot explain why only 4 out of 12 spontaneously active cells showed an IPSP in response to light. We can speculate that this is also due to the slice preparation, where excitatory inputs are cut off when DAergic inputs are spared and vice versa, but this is again pure speculation. The point of the study again is to qualitatively characterize the inhibitory response elicited by optogenetic terminal neurotransmitter release from VTA-to-LS projecting neurons. We now discuss this aspect, highlighting the point that our experiments do not allow us to conclude on the quantitative aspects of the VTA(DA)-to-LS connectivity but only examine the qualitative nature of the inhibitory response.

b - I take the authors argument that the IPSPs are likely not GABAergic, the sulpiride data are convincing. However, related to the point above, it is important to see the distribution of IPSP magnitudes elicited by VTA-LS stimulation. The authors state that 40 cells showed a response, I would like to see the distribution for these responses and statistics showing that there is significant inhibition of membrane potential across the entire population of cells that had a response.

→ We agree with the reviewer that it would be an important addition to the characterization of the VTA(DA)-to-LS input if we could quantify the distribution of IPSP magnitudes topographically. However, the nature of DA neuron inputs in the LS makes PSC size more vulnerable to slicing. Due to this limitation, as mentioned above, such analyses are not feasible. To elaborate, when the response is lacking or small, we cannot discern whether this is physiological characteristic or a slicing artifact. Considering a high presumptive 'false-negative' rate, we believe the distribution of PSC sizes would not provide meaningful physiological information. We saw more PSCs in the medial part of the 'V', and we now mention this observation in the discussion section. However, we cannot tell whether this reflects the real distribution of PSC size/incidence or is due to fewer truncated dendrites in more medially located cells. We also now include this point in the discussion section.

Also, when describing this experiment the authors state that 65% of cells show a response, n=40. I believe that this should be n=62 (with $40 = 62 \cdot 0.65$).

→ We apologize for this mistake. We revised cell numbers as follows: n = 40 cells among 61 cells

c - there are still some statements that are not backed up by appropriate quantification: "The size of the IPSPs was affected by how DA neuron projections were included in recorded slices". Where is this shown?

→ The correlation between inputs preserved in the slice and IPSC size is not directly shown. The statement is based on the subjective observation of a correlation between fluorescence intensity and size of recorded PSCs by the experimenter. We have removed the statement.

Also, "in most cases depolarizing current injection caused burst firing", where is the data for this, other than the example shown?

→ We modified this sentence to remedy the oversight: "In most cases depolarizing current injection (50 pA) caused burst firing (n = 11 cells among 14 cells), while tonic activity was observed in a minority (n = 3 cells among 14 cells)."

3 - Please include the number of animals for the RNAscope experiments, and show the variance of the quantification across animals.

→ We apologize for this oversight. We now include the number of animals and brain sections and show the variance in extended data figure 14.

Reviewer #3 (Remarks to the Author):

The authors have done a nice job in revising this manuscript. I have no further questions or comments.

→ we thank all reviewers for their time and effort!

REVIEWERS' COMMENTS

Reviewer #2 (Remarks to the Author):

The authors have addressed my remaining comments. Regarding my two major points:

- eArch3 experiments: the authors now acknowledge in the text the potential caveat of this experiment. While I remain sceptical that 55s instead of 1 min stimulation makes much of a difference in mitigating excitatory side effects, I am happy that the problems are at least acknowledged. Regarding the slice experiments, I understand that they might be hard, but in my opinion would make the paper and this conclusion more solid. In the end it's up to the authors how strongly they want to support their claims, I'm fine with the paper being published as it is.

- electrophysiology experiments: I take the potential slicing issues and agree that the negatives might be real or false. As the authors state in their reply, we don't know, and thus a qualitative analysis makes sense. But then I still fail to see the point of Fig.4d. Here the authors are providing a quantification in a handful of selected cells, to claim that the firing frequency is strongly modulated. In these cells it clearly is, but maybe across the population the effect is not significant. Maybe not because of slicing artifacts, or maybe because that is the reality. I think that this figure is misleading and should be removed. My view is further substantiated by the authors reluctance to quantify the distribution of IPSP sizes. I again take the point that the distribution might be meaningless because of potential slicing artefacts, but then so must be the data in 4d.

REVIEWER COMMENTS

REVIEWERS' COMMENTS

Reviewer #2 (Remarks to the Author):

The authors have addressed my remaining comments. Regarding my two major points:

- eArch3 experiments: the authors now acknowledge in the text the potential caveat of this experiment. While I remain sceptical that 55s instead of 1 min stimulation makes much of a difference in mitigating excitatory side effects, I am happy that the problems are at least acknowledged. Regarding the slice experiments, I understand that they might be hard, but in my opinion would make the paper and this conclusion more solid. In the end it's up to the authors how strongly they want to support their claims, I'm fine with the paper being published as it is.

→ we decided to not add any additional data.

- electrophysiology experiments: I take the potential slicing issues and agree that the negatives might be real or false. As the authors state in their reply, we don't know, and thus a qualitative analysis makes sense. But then I still fail to see the point of Fig.4d. Here the authors are providing a quantification in a handful of selected cells, to claim that the firing frequency is strongly modulated. In these cells it clearly is, but maybe across the population the effect is not significant. Maybe not because of slicing artifacts, or maybe because that is the reality. I think that this figure is misleading and should be removed. My view is further substantiated by the authors reluctance to quantify the distribution of IPSP sizes. I again take the point that the distribution might be meaningless because of potential slicing artefacts, but then so must be the data in 4d.

→ we moved the figure in question to the supplemental data.